# Which Backbone to Use: A Resource-efficient Domain Specific Comparison for Computer Vision

**Pranav Jeevan**                                               *pjeevan@iitb.ac.in*
*Department of Electrical Engineering*
*Indian Institute of Technology Bombay*

**Amit Sethi**                                                  *asethi@iitb.ac.in*
*Department of Electrical Engineering*
*Indian Institute of Technology Bombay*

**Reviewed on OpenReview:** *https://openreview.net/forum?id=XVSQnnf7QT*

## Abstract

For computer vision applications on small, niche, and proprietary datasets, fine-tuning a neural network (NN) backbone that is pre-trained on a large dataset, such as the ImageNet, is a common practice. However, it is unknown whether the backbones that perform well on large datasets, such as vision transformers, are also the right choice for fine-tuning on smaller custom datasets. The present comprehensive analysis aims to aid machine learning practitioners in selecting the most suitable backbone for their specific problem. We systematically evaluated multiple lightweight, pre-trained backbones under consistent training settings across a variety of domains spanning natural, medical, deep space, and remote sensing images. We found that even though attention-based architectures are gaining popularity, they tend to perform poorly compared to CNNs when fine-tuned on small amounts of domain-specific data. We also observed that certain CNN architectures consistently perform better than others when controlled for network size. Our findings provide actionable insights into the performance trade-offs and effectiveness of different backbones for a broad spectrum of computer vision domains.

## 1 Introduction

In computer vision, particularly for image classification, practitioners frequently employ a backbone neural network coupled with a task- and dataset-specific head. This methodology involves the backbone generating feature representations from input images, which are subsequently processed by the custom head to compute the final output. A widely adopted practice in this field is to utilize backbones pre-trained on large datasets, such as ImageNet-1k (Deng et al., 2009), followed by fine-tuning these models on smaller, domain-specific datasets. This approach is especially advantageous when specialized datasets, such as those for a particular medical application, are relatively smaller in size.

Table 1: A summary of our observations showing the top 3 backbones for fine-tuning on multiple domains for image classification

| | | | | DOMAINS | | | |
|---|---|---|---|---|---|---|---|
| Ranking | Natural | Texture | Remote Sensing | Plant | Astronomy | Medical | Overall |
| **Best** | **ConvNeXt-Tiny** | **ConvNeXt-Tiny** | **ResNeXt-50** 32 × 4d | **RegNetY-3.2GF** | **WaveMix** | **WaveMix** | **ConvNeXt-Tiny** |
| Better | EfficientNetV2-S | ResNeXt-50 32 × 4d | EfficientNetV2-S | ConvNeXt-Tiny | ConvNeXt-Tiny | EfficientNetV2-S | RegNetY-3.2GF |
| Good | RegNetY-3.2GF | RegNetY-3.2GF | ConvNeXt-Tiny | ShuffleNetV2 2.0× | DenseNet-161 | RegNetY-3.2GF | EfficientNetV2-S |

For several real computer vision problems, practitioners use off-the-shelf models from popular deep learning libraries, such as Torchvision (TorchVision-maintainers & contributors, 2016), which provide an extensive array of backbones with initial weights that are pre-trained ImageNet. These pre-trained backbone architectures, when fine-tuned on smaller domain-specific datasets, have consistently demonstrated superior performance compared to models trained from scratch. This strategy not only enhances performance but also reduces the computational resources and training time required, making it a standard practice in computer vision.

While Torchvision documentation and open-source forums list the efficacy (e.g., top-1 accuracy) of various backbone architectures on ImageNet, we explored whether these metrics translate to similar relative performance when fine-tuned on smaller niche datasets. We systematically compared various backbones to cover a variety of scenarios involving different domains. The possible discrepancy between the performance on ImageNet and niche datasets highlights the need for careful selection of backbone models for various application scenarios, and underscores the complexity and variability of transfer learning in computer vision.

Additionally, one of the major challenges for practitioners working with real-world problems is compute limitations. These limitations manifest GPU availability, power requirements, budgets, training and inference time, and even model size. To address these challenges, our experiments focus on comparing only the lightweight (model size less than 100 MB) backbone architectures available on Torchvision, which are designed to be resource-efficient with high inference speed.

The absence of benchmarks for these lightweight backbones across multiple domains poses a significant challenge for practitioners when selecting the most appropriate backbone for specific datasets with limited data. Moreover, the size of the fine-tuning dataset can influence the performance of these backbone architectures. Our study aims to investigate the impact of dataset size on the selection of backbone models and provide insights into choosing the most suitable backbone based on the specific domain and the available fine-tuning data.

We hope this analysis will assist the research community in recognizing the fundamental constraints of current architectures and help in the development of more advanced models in the future.

## 2   Related Works

The majority of research on image classification architectures leverages the ImageNet benchmark to compare the performance of different backbone models. The PyTorch Image Models (TIMM) library (Wightman, 2019) provides extensive benchmarking of ImageNet classification performance across various backbone architectures. The Visual Task Adaptation Benchmark (VTAB) (Zhai et al., 2020) evaluates the performance of computer vision models on 19 tasks spanning different domains, offering a comprehensive assessment of model adaptability.

Brigato et al. (Brigato et al., 2021) designed a benchmark focused on data-efficient image classification across various domains, demonstrating that tuning hyperparameters such as learning rate, weight decay, and batch size can create a competitive baseline that outperforms many specialized methods. Similarly, Taher et al. (Taher et al., 2021) benchmarked various self-supervised pre-training methods, evaluating 14 pre-trained ImageNet models on seven diverse medical tasks, highlighting the potential of self-supervised learning in specialized domains.

Battle of Backbones (Goldblum et al., 2023) benchmarked a selection of pre-trained ImageNet backbones, including those pre-trained via self-supervised learning and stable diffusion, across a diverse array of computer vision tasks. These tasks ranged from image classification and out-of-distribution generalization to image retrieval and object detection.

Previous research on transfer learning (Kornblith et al., 2019) has demonstrated that performance on ImageNet does not consistently align with performance on downstream tasks. Notably, adversarially-trained-models have been shown to outperform non-adversarially-trained models in transfer scenarios, particularly in data-scarce domains (Utrera et al., 2021; Salman et al., 2020), despite the latter achieving higher pre-

training accuracy. Additionally, sparse models have been found to match or even exceed the transfer learning performance of dense models (Iofinova et al., 2022), even under conditions of high sparsity.

Despite these comprehensive studies, there is a noticeable gap in the benchmarking of transfer learning performance for popular lightweight backbones across multiple domain-specific datasets. This gap indicates a need for further research to understand the transferability and efficacy of lightweight models in various application scenarios.

## 3  Backbone Architectures

Given the widespread use of off-the-shelf ImageNet pre-trained models from Torchvision library (TorchVision-maintainers & contributors, 2016), we selected models from this library for our experiments. To ensure resource efficiency, we applied specific criteria for model selection. Our primary constraint was to choose models with fewer than 30 million (M) parameters, resulting in a storage size of approximately 100 MB. Additionally, we included the WaveMix (Jeevan et al., 2024) model in our list of backbones, as it has demonstrated state-of-the-art (SOTA) performance across multiple image classification datasets from diverse domains, including EMNIST (Cohen et al., 2017) and galaxy morphology datasets.

Torchvision offers 20 backbone families, each containing models of various sizes, totaling 115 available models at the time of this writing. By applying the lightweight model constraint (fewer than 30 million parameters), we narrowed our selection to 53 models spanning 14 backbone families. From each of these families, we chose the model with the highest ImageNet top-1 accuracy for our experiments. Furthermore, we excluded models with an ImageNet top-1 accuracy below 75%, ensuring that only high-performing models were included in our evaluation.

The final list of 11 selected backbones is shown in Table 2. We included both Swin-Tiny (Liu et al., 2021) and SwinV2-Tiny (Liu et al., 2022a) in our list due to the under-representation of attention-based transformer architectures. The only other attention-based model available in Torchvision is the vision transformer (ViT), which contains 86 M parameters, far exceeding our parameter constraint. We aimed to understand how attention-based models perform in resource-efficient and low data regimes since there has been a recent trend among computer vision practitioners to use transformers for all tasks. A brief overview of the selected architectures is provided below.

**ResNet He et al. (2015)**: ResNets are the most popular and successful backbone architectures currently in use today since its arrival almost a decade ago. ResNet uses residual connections to allow for the training of very deep convolutional neural networks (CNN) by mitigating the vanishing gradient problem. We use ResNet-50 for our experiments.

**WaveMix Jeevan et al. (2024)**: WaveMix is a token-mixing architecture that uses 2-dimensional discrete wavelet transform for spacial token-mixing and has been shown to provide SOTA performance in multiple image classification datasets. We use WaveMix-192/16 (level 3) for our experiments.

**ConvNeXt Liu et al. (2022b)**: ConvNeXt is a recent CNN architecture designed to improve upon traditional CNNs by incorporating elements from transformer models, resulting in enhanced performance and scalability. It uses depth-wise convolutions, inverted bottleneck blocks and large kernels. We use ConvNeXt-Tiny in our experiments.

**Swin Transformer Liu et al. (2021)**: Swin transformer was an improvement over conventional ViT which overcame the massive data requirements for training. It incorporated efficiency using hierarchical representations, limiting the attention window and merging them stage by stage. We use Swin-Tiny and SwinV2-Tiny Liu et al. (2022a) for our experiments.

**EfficientNet Tan & Le (2021)**: EfficientNet is a family of CNNS that optimize both model size and speed by utilizing a compound scaling method that uniformly scales network depth, width, and resolution. It incorporates advanced techniques such as progressive learning and a mix of regular and mobile convolutions. We use EfficientNetV2-S in our experiments.

Table 2: List of popular NN backbones for vision that we used for in experiments for which pre-trained weights were available. All model weights were taken from Torchvision library except for WaveMix which was taken from GitHub.

| ARCHITECTURES | # PARAMS (M) | IMAGENET-1k TOP-1 ACCURACY (%) |
|---|---|---|
| ResNet-50 (He et al., 2015) | 25.6 | 76.13 |
| WaveMix-192/16 (level 3) (Jeevan et al., 2024) | 27.9 | 75.32 |
| ConvNeXt-Tiny (Liu et al., 2022b) | 28.6 | 82.52 |
| Swin-Tiny (Liu et al., 2021) | 28.3 | 81.47 |
| SwinV2-Tiny (Liu et al., 2022a) | 28.4 | 82.07 |
| EfficientNetV2-S (Tan & Le, 2021) | 21.5 | 84.23 |
| DenseNet-161 (Huang et al., 2018) | 28.7 | 77.14 |
| MobileNetV3-Large (Howard et al., 2019) | 5.5 | 75.27 |
| RegNetY-3.2GF (Radosavovic et al., 2020) | 19.4 | 81.98 |
| ResNeXt-50 $32 \times 4$d (Xie et al., 2017) | 25.0 | 81.20 |
| ShuffleNetV2 2.0× (Ma et al., 2018) | 7.4 | 76.23 |

**Densenet Huang et al. (2018)**: DenseNet is a CNN architecture that connects each layer to every other layer in a feed-forward fashion, promoting feature reuse and reduction in number of parameters. This dense connectivity pattern helps alleviate the vanishing gradient problem and leads to improved training efficiency and accuracy. We use Densenet-161 in our experiments.

**MobileNet Howard et al. (2019)**: MobileNetV3 is a CNN architecture designed for on-device and resource-constrained environments, which combines lightweight depth-wise separable convolutions with squeeze-and-excitation modules. It is a highly efficient model with improved accuracy and reduced computational complexity. We use MobileNetV3-Large in our experiments.

**RegNet Radosavovic et al. (2020)**: RegNet is a family of CNN architectures that utilize a regular design space to systematically generate a diverse range of models, optimizing for both efficiency and performance. It focuses on simple, scalable structures with uniform depth, width, and group convolution patterns, incorporating features like bottleneck blocks. We use RegNetY-3.2GF for our experiments.

**ResNeXt Xie et al. (2017)**: ResNeXt is a CNN architecture that extends the ResNet model by introducing a cardinality dimension, using grouped convolutions to aggregate multiple transformations, which improves performance and efficiency. We use ResNeXt-50 $32 \times 4$d for our experiments.

**ShuffleNet Ma et al. (2018)**: ShuffleNet is a lightweight CNN architecture designed for efficient computation on mobile devices. It uses channel shuffling and point-wise group convolution to optimize speed and accuracy. We use ShuffleNetV2 2.0× for our experiments.

## 4 Datasets

To evaluate the performance of various backbones on data-efficient fine-tuning, we decided to conduct our experiments on image classification datasets with a maximum of 100,000 training images. We selected 20 publicly available datasets from seven different domains: natural images, textures, remote sensing, plants, astronomy, surface defects, and medical imaging, with number of training images ranging from 1000 to 100,000 and number of classes ranging from two to 200. Details of the datasets used in each domain are provided in Table 3 and the supplementary materials.

Table 3: Details of image classification datasets used, their domain details, the number of images in training and testing sets, and the number of classes.

| DATASETS | DOMAIN DESCRIPTION | # TRAINING IMAGES | # TESTING IMAGES | # CLASSES |
|---|---|---|---|---|
| CIFAR-10 (Krizhevsky, 2009) | Natural Images | 50,000 | 10,000 | 10 |
| CIFAR-100 (Krizhevsky, 2009) | Natural Images | 50,000 | 10,000 | 100 |
| Tiny ImageNet (Le & Yang, 2015) | Natural Images (ImageNet subset) | 100,000 | 10,000 | 200 |
| Stanford dogs (Khosla et al., 2011) | Natural Images (Dog breeds) | 12,000 | 8,580 | 120 |
| Flowers-102 (Nilsback & Zisserman, 2008) | Natural Images (Flower species) | 2,040 | 6,149 | 102 |
| CUB-200-2011 (Welinder et al., 2010) | Natural Images (Bird species) | 5,994 | 5,794 | 200 |
| Stanford Cars (Dehghan et al., 2017) | Natural Images (Car models) | 8,144 | 8,041 | 196 |
| Food-101 (Bossard et al., 2014) | Natural Images (Food categories) | 75,750 | 25,250 | 101 |
| DTD (Cimpoi et al., 2014) | Texture Images | 1,880 | 1,880 | 47 |
| NEU Surface Defects (Song & Yan, 2013) | Surface Defect Images | 1,440 | 360 | 6 |
| UC Merced Land Use (Yang & Newsam, 2010) | Remote Sensing Images | 1,680 | 420 | 21 |
| EuroSAT (Helber et al., 2019) | Remote Sensing Images | 18,900 | 8,100 | 10 |
| PlantVillage (Hughes & Salathe, 2016) | Plant Images | 44,343 | 11,105 | 39 |
| PlantCLEF (Goëau et al., 2021) | Plant Images | 10,455 | 1135 | 20 |
| Galaxy10 DECals (Leung & Bovy, 2018) | Astronomy Images (Galaxy Morphology) | 15,962 | 1,774 | 10 |
| BreakHis 40× (Spanhol et al., 2016) | Medical Images (Histopathology) | 1,398 | 606 | 2 |
| BreakHis 100× (Spanhol et al., 2016) | Medical Images (Histopathology) | 1,458 | 632 | 2 |
| BreakHis 200× (Spanhol et al., 2016) | Medical Images (Histopathology) | 1,411 | 611 | 2 |
| BreakHis 400× (Spanhol et al., 2016) | Medical Images (Histopathology) | 1,276 | 553 | 2 |
| RSNA Pneumonia Detection (Stein et al., 2018) | Medical Images (Radiology) | 24,181 | 6046 | 2 |

## 5 Experimental Details

To measure the fine-tuning performance of the selected backbones in image classification via transfer learning, we employed a standard training protocol to ensure a fair comparison across all models[1]. In our experiments, we did not freeze any layers of the pre-trained backbones, opting instead to fine-tune the entire model on each dataset or a fraction of it. For each dataset, the only modification made to the backbone was in the final classification layer, which was adjusted to match the number of classes specific to the dataset. The performance was measured using top-1 accuracy on the test set.

All images were resized to $256 \times 256$ for our experiments, except for BreakHis, where we resized images to $672 \times 448$ since reducing the resolution of histopathology images leads to poor results across models.

TrivialAugment (Müller & Hutter, 2021) was used as data augmentation for all datasets except BreakHis and Galaxy10 DECals. Augmentation was only applied after verifying that its usage improved model performance.

We used early stopping, halting training if accuracy did not improve after 10 epochs. All experiments were done on a single 80 GB Nvidia A100 GPU. We employed DualOpt (Jeevan & sethi, 2022) for training, starting with the AdamW optimizer ($\alpha = 0.001, \beta_1 = 0.9, \beta_2 = 0.999, \epsilon = 10^{-8}$) with a weight decay of 0.01 during the initial phase and used SGD with a learning rate of 0.001 and momentum = 0.9 during the final phase. No attempt was made to tune learning rates for each model. We did not use any learning rate schedulers or label-smoothening.

Cross-entropy loss was used for fine-tuning the models. The batch size was chosen to maximize GPU utilization during training. We used automatic mixed precision in PyTorch during training. Top-1 accuracy on the test set for best of three runs with random initialization is reported as a generalization metric based on prevailing protocols (Hassani et al., 2021).

---

[1]Our training code is available at `https://github.com/pranavphoenix/Backbones`

Table 4: Top-1 classification accuracy for fine-tuning pre-trained backbones on natural image datasets. The top three results for each dataset are highlighted with best in green and bold, second best in lighter green, and third best in lightest green.

| BACKBONES | Stanford Dogs | Flowers102 | CUB200 | NATURAL IMAGES Stanford Cars | Tiny ImageNet | CIFAR10 | CIFAR100 | Food101 |
|---|---|---|---|---|---|---|---|---|
| ResNet-50 | 82.03 | 90.77 | 75.34 | 90.84 | 75.33 | 94.58 | 79.05 | 88.36 |
| WaveMix | 83.12 | 88.43 | 67.32 | 87.86 | 76.30 | **97.26** | **83.88** | 85.88 |
| ConvNeXt-Tiny | **89.47** | **94.41** | **81.92** | **92.26** | **83.42** | 96.48 | 82.60 | 89.43 |
| Swin-Tiny | 83.81 | 90.07 | 78.16 | 90.04 | 76.85 | 95.38 | 81.25 | 88.81 |
| SwinV2-Tiny | 84.44 | 90.06 | 66.89 | 91.38 | 73.93 | 94.52 | 75.37 | 89.04 |
| EfficientNetV2-S | 86.59 | 93.65 | 79.10 | 91.59 | 81.35 | 96.42 | 83.20 | 89.84 |
| DenseNet-161 | 80.02 | 89.96 | 76.44 | 91.06 | 75.03 | 96.03 | 81.01 | 87.02 |
| MobileNetV3-Large | 78.85 | 89.36 | 73.91 | 85.50 | 76.74 | 95.92 | 79.51 | 85.73 |
| RegNetY-3.2GF | 85.94 | 92.22 | 81.24 | 91.33 | 80.14 | 96.82 | 82.89 | **90.40** |
| ResNeXt-50 32×4d | 85.09 | 91.78 | 78.29 | 90.22 | 77.86 | 96.16 | 81.30 | 88.89 |
| ShuffleNetV2 2.0× | 78.11 | 91.54 | 74.73 | 88.46 | 76.43 | 96.30 | 83.32 | 86.21 |

Table 5: Top-1 classification accuracy for fine-tuning pre-trained backbones on different domain datasets. The top three results are highlighted with best in green and bold, light green, and third best in lightest green.

| DOMAIN | TEXTURE | REMOTE SENSING | | PLANT | | ASTRONOMY | SURFACE DEFECTS | MEDICAL (RADIOLOGY) |
|---|---|---|---|---|---|---|---|---|
| Backbones | DTD | UC Merced Land Use | Eurosat | PlantVillage | PlantCLEF | Galaxy | NEU | RSNA |
| ResNet-50 | 68.21 | 96.90 | 98.75 | 99.83 | 80.60 | 84.80 | 99.72 | 87.30 |
| WaveMix | 68.25 | 97.72 | **98.96** | 99.79 | 79.52 | **95.10** | **100.00** | 86.94 |
| ConvNeXt-Tiny | **73.70** | 98.33 | 98.74 | 99.80 | 82.71 | 87.29 | **100.00** | 86.71 |
| Swin-Tiny | 70.15 | 97.86 | 98.52 | 99.70 | 79.03 | 84.64 | 95.28 | 70.36 |
| SwinV2-Tiny | 68.78 | **98.81** | 98.50 | 99.76 | 78.16 | 83.15 | 99.72 | 86.38 |
| EfficientNetV2-S | 70.18 | 98.22 | 98.88 | 99.81 | 81.09 | 84.75 | **100.00** | 87.06 |
| DenseNet-161 | 66.14 | 97.08 | 98.83 | **99.88** | 76.49 | 86.70 | 97.07 | 87.10 |
| MobileNetV3-Large | 68.69 | 97.14 | 98.72 | 99.80 | 75.52 | 82.47 | 61.39 | 87.05 |
| RegnetY-3.2GF | 71.16 | 98.33 | 98.69 | 99.83 | **82.76** | 82.47 | **100.00** | 87.05 |
| ResNeXt-50 32×4d | 72.73 | 98.33 | **98.96** | 99.75 | 80.32 | 85.82 | **100.00** | **87.89** |
| ShuffleNetV2 2.0× | 68.64 | 97.86 | 98.65 | 99.70 | 81.76 | 83.69 | 86.39 | 87.53 |

Table 6: Top-1 classification accuracy for fine-tuning pre-trained backbones on medical datasets. The top three results are highlighted with best in green and bold, light green, and third best in lightest green.

| Backbones | MEDICAL (HISTOPATHOLOGY) BREAKHIS DATASET 40× | 100× | 200× | 400× | Average |
|---|---|---|---|---|---|
| ResNet-50 | 97.91 | **99.53** | 99.22 | 98.44 | 98.78 |
| WaveMix | 99.42 | 99.42 | 99.38 | **99.35** | **99.39** |
| ConvNeXt-Tiny | 95.10 | 90.73 | 88.59 | 88.72 | 90.79 |
| Swin-Tiny | 88.11 | 93.89 | 90.00 | 82.61 | 88.65 |
| SwinV2-Tiny | 89.96 | 92.44 | 88.65 | 83.83 | 88.72 |
| EfficientNetV2-S | 99.44 | 99.42 | 99.11 | 99.06 | 99.26 |
| DenseNet-161 | 98.62 | 99.24 | 99.28 | 98.20 | 98.84 |
| MobileNetV3-Large | 99.56 | 98.24 | **99.50** | 98.94 | 99.06 |
| RegNetY-3.2GF | **99.84** | 99.22 | 99.48 | 98.02 | 99.14 |
| ResNeXt-50 32 × 4d | 99.46 | 99.22 | 99.34 | 98.05 | 99.02 |
| ShuffleNetV2 2.0× | 99.38 | 98.59 | 99.22 | 98.02 | 98.80 |

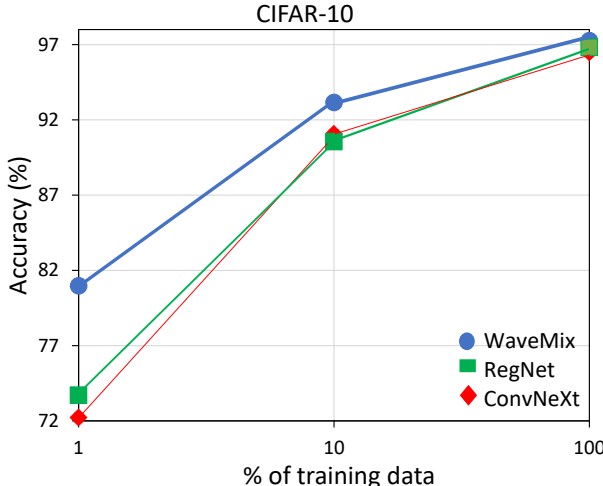

Figure 1: Accuracy of top three backbones on CIFAR-10 with different amounts of training data – 1% (500 images), 10% (5000 images) and 100% (50,000 images).

## 6 Results

### 6.1 Fine-tuning Performance

The results of our fine-tuning experiments on datasets of all domains are shown in Table 4, Table 5 and Table 6. We find that ConvNeXt-Tiny outperforms all the other models in almost all natural image datasets (except CIFAR-10 and CIFAR-100). EfficientNetV2-S also performs well on atleast six out of the eight natural image datasets. RegNetY-3.2GF also performs well on seven of the eight natural image datasets. WaveMix performs the best in CIFAR-10 and CIFAR-100 but could not replicate the same performance on the other natural image datasets.

ConvNeXt retains the good performance even among other domains such as textures, plant, surface defects, remote sensing and astronomy. RegNet also performs well on texture, remote sensing, surface defects and plant domains. We observe that SwinV2-Tiny performs the best in UC Merced Land Use dataset, WaveMix and ResNeXt top the EuroSAT dataset, ResNeXt tops RSNA dataset, DenseNet tops the PlantVillage dataset and WaveMix significantly outperforms all the other models in Galaxy10 DECals dataset.

### 6.2 Performance with Limited Training Data

We observe from Table 7 that the models which performed really well at 100% of the training data also performed better than the other models at 10% and 1% of the training data. We can see from CIFAR-10 and Tiny Imaginary Dataset results that ConvNeXt, WaveMix, EfficientNet, and even RegNet are actually performing better than others even when the training data is significantly reduced. This shows that the fine-tuning superiority of these models is relatively insensitive to the amount of training data in the range tested.

Figure 1 and Figure 2 shows the performance of top three models with training data. We see that the difference in accuracy among the three models is large and significant when the training data is less (1%) and this difference reduces as the training data increases.

We observe similar behavior for other domains in Table 8. In plant datasets, the top performing models at 100% of training data, such as ConvNeXt, EfficientNet, DenseNet, and RegNet also performed better than the other models at 10% of the training data. Similar behavior is also observed in radiology images, where ResNeXt and ResNeXt performed better than the others.

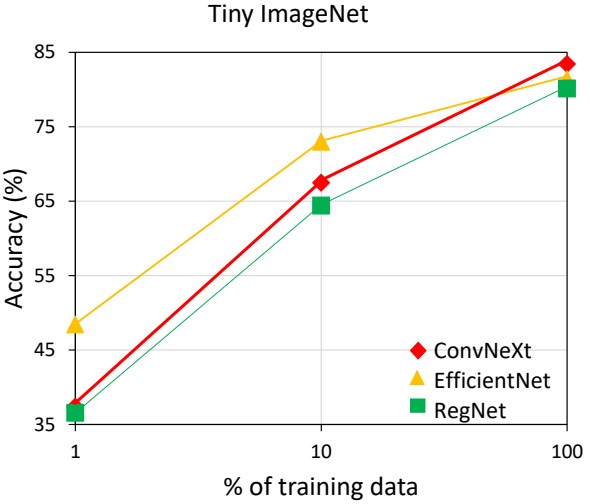

Figure 2: Accuracy of top three backbones on TinyImageNet with different amounts of training data – 1% (500 images), 10% (5000 images) and 100% (50,000 images)..

Table 7: The variation of accuracy of backbones with reduction of training data from 100% to 10% and to 1% – for CIFAR-10 and Tiny ImageNet datasets.

| | CIFAR-10 | | | TINY IMAGENET | | |
|---|---|---|---|---|---|---|
| # training images | 50,000 | 5,000 | 500 | 100,000 | 10,000 | 1,000 |
| % training data | 100% | 10% | 1% | 100% | 10% | 1% |
| | | | | | | |
| ResNet | 94.58 | 86.5 | 65.42 | 76.30 | 66.56 | 28.11 |
| WaveMix | **97.26** | **93.16** | **80.98** | 75.33 | 55.15 | 25.24 |
| ConvNeXt | 96.48 | 91.06 | 72.22 | **83.42** | 67.47 | 37.38 |
| Swin | 95.38 | 85.64 | 39.77 | 76.85 | 48.50 | 25.62 |
| SwinV2 | 94.52 | 85.66 | 46.51 | 73.93 | 39.31 | 12.29 |
| EfficientNetV2 | 96.42 | 89.44 | 77.06 | 81.35 | **72.97** | **48.42** |
| DenseNet | 96.03 | 86.92 | 64.83 | 75.03 | 51.89 | 18.00 |
| MobileNetV3 | 95.92 | 88.85 | 61.27 | 76.74 | 62.70 | 31.08 |
| RegNetY | 96.82 | 90.55 | 73.71 | 80.14 | 64.40 | 36.56 |
| ResNeXt | 96.16 | 90.24 | 73.12 | 77.86 | 67.01 | 34.40 |
| ShuffleNetV2 | 96.30 | 90.62 | 77.45 | 76.43 | 61.65 | 29.89 |

Table 8: Variation of accuracy of backbones with reduction of training data – from 100% to 10%. The top three results are highlighted with best in green and bold, light green, and third best in lightest green.

| | PLANTVILLAGE | | PLANTCLEF | | RSNA | | FOOD-101 | | EUROSAT | |
|---|---|---|---|---|---|---|---|---|---|---|
| Number of training images | 44,343 | 4,434 | 10,455 | 1,045 | 24,181 | 2,418 | 75,750 | 7,575 | 18.900 | 1,890 |
| Percentage of training data | 100% | 10% | 100% | 10% | 100% | 10% | 100% | 10% | 100% | 10% |
| | | | | | | | | | | |
| ResNet | 99.83 | 98.07 | 80.60 | 57.47 | 87.3 | 81.26 | 88.36 | 74.26 | 98.75 | 96.49 |
| WaveMix | 99.79 | 97.39 | 79.52 | 57.64 | 86.94 | 80.93 | 85.88 | 68.86 | 98.96 | 96.48 |
| ConvNeXt | 99.80 | **98.92** | 82.71 | **64.68** | 86.71 | 80.03 | 89.43 | **78.47** | 98.74 | **97.29** |
| Swin | 99.70 | 97.75 | 79.03 | 48.57 | 70.36 | 68.22 | 88.81 | 76.21 | 98.52 | 95.95 |
| SwinV2 | 99.76 | 97.72 | 78.16 | 21.47 | 86.38 | 67.63 | 89.04 | 74.17 | 98.50 | 96.48 |
| EfficientNet | 99.81 | 98.42 | 81.09 | 64.4 | 87.06 | 79.96 | 89.84 | 76.68 | 98.88 | 97.23 |
| DenseNet | **99.88** | 98.57 | 76.49 | 58.59 | 87.1 | **81.51** | 87.02 | 72.44 | 98.83 | 96.86 |
| MobileNet | 99.80 | 98.29 | 75.52 | 54.93 | 87.05 | 79.55 | 85.73 | 71.12 | 98.72 | 96.35 |
| RegNet | 99.83 | 98.25 | **82.76** | 64.53 | 87.05 | 79.56 | **90.4** | 77.89 | 98.69 | 96.49 |
| ResNeXt | 99.75 | 98.01 | 80.32 | 63.61 | **87.89** | 81.42 | 88.89 | 74.69 | **98.96** | 96.53 |
| ShuffleNet | 99.70 | 98.30 | 81.76 | 58.69 | 87.53 | 80.42 | 86.21 | 70.97 | 98.65 | 96.00 |

# 7 Observations

■ **Higher pre-training accuracy on ImageNet does not translate to higher fine-tuning accuracy.** From Table 2 we can see that among all the backbones we used for our experiments, EfficientNetV2-S has the highest pre-training ImageNet-1k accuracy (84.23%). Even though the fine-tuning performance of EfficientNet was high, it did not perform the best in any of the domains or datasets on which we evaluated the models. Even among datasets whose images are sourced from ImageNet dataset, such as Tiny ImageNet and Stanford Dogs, we find that ConvNeXt outperforms EfficientNet. Therefore, we recommend the practitioners to not use the pre-training accuracy as an ironclad criterion to choose the backbone.

■ **Convolutional models strongly outperform transformers for resource-efficient low-data fine-tuning tasks.** Even though Swin transformer has a hierarchical structure capable of exploiting the spacial inductive bias (Goldblum et al., 2023), it still performed poorly in almost all our tasks compared to modern CNN architectures such as ConvNeXt (which was designed based on macro architectural insights from the Swin transformer). Hence, we recommend that for fine-tuning on small datasets, it is better to avoid using transformer architectures, such as Swin, and use pure CNN backbones, such as ConvNeXt, EfficientNet or RegNet.

■ **ConvNeXt architecture consistently outperforms other models when fine-tuning on natural image datasets.** This superior performance can be attributed to the thought design of ConvNeXt, which integrates architectural advancements that bridge the gap between traditional convolutional networks, such as ResNet, and modern Swin transformer models. ConvNeXt retains the beneficial convolutional inductive bias, enabling it to learn more effectively than attention-based transformer models. Our data clearly indicates that for natural images, ConvNeXt stands out as the best model, delivering exceptional fine-tuning performance.

■ **RegNet and EfficientNet models are excellent choices for fine-tuning across a wide range of image domains.** While ConvNeXt excels predominantly with natural images, EfficientNet closely follows in performance, and RegNet also shows strong results in this domain. However, the versatility of RegNet and EfficientNet extends beyond natural images. Our experiments reveal that these models also perform exceptionally well on diverse domains, including remote sensing images, plant datasets, and medical images, such as histopathology images. Therefore, we recommend practitioners to consider RegNet and EfficientNet when working with datasets beyond natural images, as their adaptability and robust performance across various domains make them valuable tools for fine-tuning tasks.

■ **ShuffleNet is a better choice than MobileNet when very light-weight models are needed.** Among very lightweight models, specifically those with a model size of less than 50 MB, which are ideal for on-device applications, we find that ShuffleNetV2 generally outperforms MobileNetV3 across multiple domains. Although MobileNetV3 has shown better performance on medical domain, ShuffleNetV2 demon-

strates more consistent and slightly superior performance across a broader range of image domains. Therefore, we recommend ShuffleNetV2 as a better choice for practitioners dealing with on-device applications, where fine-tuning a model on a domain-specific dataset is required.

■ **WaveMix performs well in datasets where multi-resolution token-mixing aids in learning** WaveMix outperforms all other models (9% increase from the second best, ConvNeXt) in galaxy morphology classification. WaveMix also performs well in medical domain performing better than all other datasets, and also maintaining the performance across different magnification. WaveMix, which uses 2D-DWT might possess inductive bias that can analyse the domains of astronomy and medical images better than other convolutional models due its multi-resolution token-mixing. Multiple levels of 2D-DWT also gives more significance to low frequency components (shapes) compared to regular convolutions which are biased towards higher frequency features such as textures. We recommend using WaveMx in domains where features across different resolutions are needed for better performance. WaveMix also performs better than ConvNeXt in CIFAR-10 and CIFAR-100 datasets, which were actually low resolution natural images ($32 \times 32$) which were resized (to $256 \times 256$) for our training. Similarly, it gives best performance in remote-sensing dataset, EuroSAT, whose images ($64 \times 64$) were resized (to $256 \times 256$) for our training. WaveMix is also state-of-the-art in many low-resolution datasets such as EMNIST ($28 \times 28$). The only low resolution image dataset where WaveMix did not perform well is Tiny ImageNet whose images ($64 \times 64$) were also resized (to $256 \times 256$). We attribute this to the fact that Tiny ImageNet is an ImageNet-1k subset and the other models which has better performance in ImageNet-1k naturally performed better. So, we recommend using WaveMix for low-resolution image datasets.

■ **Age of ResNet dominance is over.** Our results also reveal that ResNet is no longer competitive compared to these modern architectures in any domain. Our experimental data shows that for natural images, ResNet does not even rank among the top three performers. It significantly lags behind newer models, such as ConvNeXt. While ResNet does perform relatively well on one medical dataset, even in this case, other models achieve similar performance levels. Therefore, we recommend that practitioners should transition from using ResNet to these newer architectures while acknowledging the tremendous contributions of ResNet in inspiring several of the the later architectures.

■ **ConvNeXt and Swin transformers perform poorly in medical domain.** Our results indicate that both of these architectures perform significantly worse compared to other models when used in histopathology and radiology datasets in medical domain. Therefore, we advise practitioners in the medical field to be careful while using these models, despite their higher performance in other domains.

■ **Most of the top models in every domain retain their higher performance even with less training data.** We find that even when we fine-tune with a small percentage of training data (even 1% $\sim 1000$ images), the models which performed well with full training set still retained their superiority. This points to the presence of a domain specific inductive bias present in these models since they can learn better representations with very less data. The failure of these models to perform well on other domains similarly with less training data also alludes to this inductive bias.

## 8 Architectural Discussions

All top-performing models across various domains are the latest architectures, such as RegNet, EfficientNet, WaveMix, and ConvNeXt. The Table 9 displays the architectural details of all the convolutional architectures used in our experiments, arranged from oldest to newest. We observe that certain architectural trends have improved the performance of the newer models compared to older ones. These observations can be leveraged to develop more resource-efficient backbones for fine-tuning tasks.

Most of the models use a $1 \times 1$, $3 \times 3$, $1 \times 1$ kernel structure in their blocks, with a normalisation and activation following each convolutional operation. MobileNet employs $5 \times 5$ kernels in the final stages, while ConvNeXt uses a $7 \times 7$ kernel for spatial token-mixing in each block. All architectures use $1 \times 1$ convolution to change the number of channels.

While changing the channel dimensions inside bottlenecks or inverted bottlenecks, nearly all models employ a channel expansion or contraction factor of 4. MobileNet and EfficientNet use even higher values, up to

Table 9: Architectural comparison of the convolutional backbones showing the block structure of the residual block (eg. BottleNeck (BN)), kernels used in each block in sequence, number of stages in pyramidal models, number of blocks in each stage in pyramidal models, the non-linearities used in networks blocks, the channel expansion factor (CEF) in each block, the kernel and stride used in the initial stem layer, whether depth-wise convolutions (DConv) and squeeze-excitation (SE) operations were used in blocks, and the normalization used in blocks (whether BatchNorm (BN) or LayerNorm (LN)). The architectures are listed in order from oldest to newest.

| Backbone | Block | Kernels in Block | # Stages | #Blocks/stage | Non-linearity | CEF | Stem | DConv | SE | Norm |
|---|---|---|---|---|---|---|---|---|---|---|
| ResNet-50 | Bottleneck | 1×1, 3×3, 1×1 | 4 | 3,4,6,3 | ReLU | 4 | 7×7, /2 | No | No | BN |
| DenseNet-161 | Inverted BN | 1×1, 3×3 | 4 | 6,12,36,24 | ReLU | 4 | 7×7, /2 | No | No | BN |
| ResNeXt-50 32×4d | Bottleneck | 1×1, 3×3, 1×1 | 4 | 3,4,6,3 | ReLU | 2 | 7×7, /2 | No | No | BN |
| ShuffleNetV2 2.0× | Isotropic | 1×1, 3×3, 1×1 | 3 | 4,8,4 | ReLU | 1 | 3×3, /2 | Yes | No | BN |
| MobileNetV3-Large | Inverted BN | 1×1, 3×3/5×5, 1×1 | 4 | 2,3,6,3 | ReLU, SiLU | 3-6 | 3×3, /2 | Yes | Yes | BN |
| RegNetY-3.2GF | Isotropic | 1×1, 3×3, 1×1 | 4 | 2, 5, 13, 1 | ReLU | 1 | 3×3, /2 | No | Yes | BN |
| EfficientNetV2-S | Inverted BN | 1×1, 3×3, 1×1 | 6 | 2, 4, 4, 6, 9, 15 | SiLU | 4-6 | 3×3, /2 | Yes | Yes | BN |
| WaveMix-192/16 | Tokenmixer | 1×1, 1×1, 1×1 | 1 | 16 | GELU | 2 | 4×4, /4 | No | No | BN |
| ConvNeXt-Tiny | Tokenmixer | 7×7, 1×1, 1×1 | 4 | 3,3,9,3 | GELU | 4 | 4×4, /4 | Yes | No | LN |

6. All models utilize adaptive average pooling in the classification head to down-sample the output feature maps from the final stage to a $1 \times 1$ spatial resolution before passing it through the final linear layer to get class probabilities.

■ **Inverted bottleneck design is better.** The older models used bottleneck blocks where channel dimension was reduced using $1 \times 1$ convolutions before being passed to the main $3 \times 3$ convolutions to increase efficiency. Newer architectures such as EfficientNet and MobileNet use an inverted bottleneck structure where the $1 \times 1$ convolutions increases the channel dimension before $3 \times 3$ convolutions. The increase in parameters is offset mainly by using the parameter efficient depth-wise convolution which also increases performance. Models, such as ShuffleNet and RegNet keeps the channel dimension invariant throughout the block resulting in an isotropic design. WaveMix and ConvNeXt follow the token-mixing structure where spatial and channel mixing is performed separately, with spacial mixing done with token-mixing operations such as wavelet transform and large kernel depth-wise convolution respectively. The channel dimension is expanded only for channel mixing using $1 \times 1$ convolutions in an inverted bottleneck design.

■ **Depth-wise convolution replacing regular convolution.** Depth-wise convolutions have fewer parameters and requires less operations than regular convolutions. Most of the newer models such as ShuffleNet, MobileNet and EfficientNet replace the $3 \times 3$ kernel regular convolution with $3 \times 3$ kernel depth-wise convolution. ConvNeXt uses $7 \times 7$ kernel depth-wise convolution for efficient spacial token-mixing. All $1 \times 1$ convolutions in all these models are point-wise convolutions (regular convolutions). SiLU activation has been observed to give better performance when used with depth-wise convolutions (Radosavovic et al., 2020).

■ **Four-stage models with largest number of blocks in penultimate stage.** From the oldest to the newest models, a four-stage structure has been the most popular and has consistently resulted in good performance. Exceptions to this include EfficientNet with six stages, ShuffleNet with 3 stages, and WaveMix with a single stage; all other backbones have four stages. A common design principle observed in most models is the increasing number of blocks with each subsequent stage, except for the final stage. Only EfficientNet has more blocks in the final stage than in the penultimate stage. Additionally, most four-stage models use a channel expansion factor greater than or equal to two during feature resolution down-sampling between stages to increase the number of channels.

■ **Strided convolution for down-sampling.** With the exception of DenseNet which used average pooling for down-sampling the feature resolution between different stages of the network, all other convolutional models used strided convolutions with stride two for feature resolution down-sampling. So strided convolutions are better for model performance than pooling operations.

■ **Small kernel stem layer.** Stem layer is the initial layer used to down-sample the full size input image whose output is then sent to the body of architectures which has multiple stages. Compared to older models which used large $7 \times 7$ kernel with stride 2, newer models use $3 \times 3$ and $4 \times 4$ kernels. The token-mixers use patchify stem ($4 \times 4$ non-overlapping convolutions).

■ **SiLU and GELU replacing ReLU.** Older models used ReLU activation after convolution operations. The newer models use a combination of ReLU and SiLU activations while the token-mixers inspired from transformer architecture uses GELU activation. Compared to other convolutional models which uses 2 to 3 activations per block, token-mixers use much less activations (one per block).

■ **Addition of squeeze-excitation (SE) module. (Hu et al., 2019)** SE module focus on the most informative features through explicit modeling of channel inter-dependencies and can be easily integrated into various architectures without significant changes. The top performing architectures, such as RegNet, EfficientNet, used SE modules.

## 9   Conclusions

In the present study, we have compared the performance of various lightweight, resource-efficient backbones that were pre-trained on ImageNet across different domains (including medical, natural, astronomy, plant, and remote sensing images) when fine-tuned for performance. Our analysis revealed that modern architectural models, such as ConvNeXt, EfficientNet, and RegNet excel in handling multiple domains images. Additionally, models with specific inductive biases, such as WaveMix, are particularly useful for tasks requiring multi-resolution analysis. Among lightweight, on-device models, ShuffleNet consistently outperformed MobileNet in fine-tuning tasks. We also observed that transformer-based or attention-based models, such as Swin Transformer, do not perform well when the fine-tuning dataset is small.

Based on our findings, we offer practical recommendations for using pre-trained computer vision backbones, which can be a valuable guide for practitioners and researchers alike, who aim to optimize their models for various image domains. We hope that our work will contribute to the development of better model architectures capable of performing well across diverse image datasets.

**Limitations:** We restricted our comparison to models available in Torchvision, focusing specifically on lightweight and resource-efficient architectures. Consequently, we did not analyze any larger models with more than 30 million parameters, limiting our ability to test the scalability of these models with larger fine-tuning datasets. Furthermore, our analysis was confined to fine-tuning datasets containing fewer than 100,000 images, which may not fully represent scenarios involving significantly larger datasets and scalability of these backbones.

Another limitation of our work is that we exclusively focused on the computational task of image classification. We did not extend our analysis to other important tasks in computer vision, such as object detection or image retrieval. The performance of various backbones on these other tasks remains unexplored in our study. While we hope that there might be some correlation with the performance we observed to these other computer vision tasks, this remains speculative and requires further investigation to confirm.

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

# A    Appendix

## A.1    Dataset Details

### A.1.1    Natural image datasets

**CIFAR-10 Krizhevsky (2009)**: CIFAR-10 dataset is a widely-used benchmark dataset for image classification, consisting of 60,000 $32 \times 32$ color images in 10 different classes, with 6,000 images per class divided into 50,000 training images and 10,000 test images.

**CIFAR-100 Krizhevsky (2009)**: CIFAR-100 dataset uses the same images as CIFAR-10, but images are distributed across 100 different classes, with 600 images per class divided into 50,000 training images and 10,000 test images. It provides a more challenging classification task compared to CIFAR-10 due to the larger number of classes.

**Tiny ImageNet Le & Yang (2015)**: Tiny ImageNet dataset is a subset of the ImageNet dataset, consisting of 200 image classes with 500 training images and 50 test images per class, each resized to $64 \times 64$ pixels. It is widely used for benchmarking image classification algorithms, particularly in low-resource scenarios.

**Stanford Dogs Khosla et al. (2011)**: Stanford Dogs dataset is a comprehensive dataset for fine-grained image classification, containing 20,580 images of 120 different dog breeds. It is widely used for benchmarking algorithms, particularly in distinguishing between closely related categories.

**Flowers-102 Nilsback & Zisserman (2008)**: The Flowers 102 dataset is a dataset for fine-grained image classification, consisting of 8,189 images of flowers categorized into 102 different species. Each class has between 40 to 258 images, and the dataset is commonly used to benchmark algorithms in classification tasks due to its diversity and challenging nature.

**CUB-200-2011 Welinder et al. (2010)**: Caltech-UCSD Birds-200-2011 is a comprehensive dataset for fine-grained image classification, consisting of 200 bird species with 11,788 annotated images. It is widely used for benchmarking algorithms in fine-grained visual recognition tasks due to its high level of granularity.

**Stanford Cars Dehghan et al. (2017)**: The Stanford Cars dataset is a large-scale dataset for fine-grained image classification, consisting of 16,185 images of 196 classes of cars. It is widely used for evaluating and benchmarking computer vision algorithms in tasks involving fine-grained visual recognition and object detection.

**Food-101 Bossard et al. (2014)**: Food-101 dataset is a large-scale dataset for food classification, containing 101,000 images of 101 different food categories, with 750 training images and 250 test images per class. It is commonly used to benchmark image recognition algorithms in the context of food and culinary applications.

### A.1.2    Texture image dataset

**DTD Cimpoi et al. (2014)**: Describable Textures Dataset (DTD) is a collection of 5,640 texture images categorized into 47 classes based on human-describable attributes. It is used to evaluate and benchmark algorithms in texture recognition and classification tasks.

### A.1.3   Surface Defect dataset

**NEU Surface Defect Song & Yan (2013)**: The Northeastern University (NEU) surface defect database is a benchmark dataset for surface defect detection and classification, featuring images of six types of surface defects. The dataset includes 1,800 images with a resolution of $200 \times 200$ pixels, divided into 1,200 training images and 600 testing images. Each defect type has an equal number of images, making it a well-balanced dataset for training and evaluating machine learning models.

### A.1.4   Remote Sensing datasets

**UC Merced Land Use Yang & Newsam (2010)**: UC Merced Land Use dataset is a high-resolution dataset for land use classification, containing 2,100 aerial images categorized into 21 land use classes with 100 images per class. Each image is $256 \times 256$ pixels, and the dataset is commonly used for evaluating and benchmarking algorithms in remote sensing and geospatial analysis tasks.

**EuroSAT Helber et al. (2019)**: The EuroSAT dataset is a benchmark dataset for land use and land cover classification, consisting of 27,000 RGB and multi-spectral images covering 10 classes, with images derived from Sentinel-2 satellite data. It is widely used for evaluating the performance of machine learning algorithms in remote sensing and geospatial analysis tasks.

### A.1.5   Plant image datasets

**PlantVillage Hughes & Salathe (2016)**: PlantVillage dataset is a comprehensive dataset for plant disease classification, containing over 54,000 images of healthy and diseased leaves across 39 different plant categories. It is widely used for benchmarking machine learning algorithms in agricultural and plant pathology applications.

**PlantCLEF Goëau et al. (2021)**: The PlantCLEF dataset is a large-scale dataset for plant identification, comprising millions of images covering thousands of plant species, including trees, flowers, fruits, and leaves. It is used for evaluating and benchmarking algorithms in botanical classification and plant biodiversity studies. We use a subset of this dataset.

### A.1.6   Astronomy dataset

**Galaxy 10 DECals Leung & Bovy (2018)**: Galaxy 10 DECals dataset is a dataset for galaxy classification, consisting of 17,000 images of galaxies classified into 10 different morphological categories. It is used for evaluating and benchmarking machine learning algorithms in astronomy and astrophysical research.

### A.1.7   Medical image datasets

**BreakHis Spanhol et al. (2016)**: Breast Cancer Histopathology Database is a dataset specifically designed for the classification of breast cancer histopathological images. It contains 7,909 microscopic images of breast tumor tissue, divided into benign and malignant categories. It provides microscopic images of breast tumor tissue at four different magnification levels: $40\times$, $100\times$, $200\times$ and $400\times$. Each magnification level offers a different level of detail, allowing for a comprehensive analysis of histopathological features. The dataset is widely used for evaluating and bench-marking algorithms in medical image analysis and computer-aided diagnosis.

**RSNA Pneumonia Detection Challenge Dataset Stein et al. (2018)**: The RSNA Pneumonia Detection Challenge dataset, provided by the Radiological Society of North America, consists of over 30,000 chest X-ray images annotated for the presence of pneumonia. It is designed to facilitate the development of machine learning models for pneumonia detection, promoting advancements in medical imaging analysis.

