# OpenReview forum: "Which Backbone to Use: A Resource-efficient Domain Specific Comparison for Computer Vision"
_TMLR — Accepted by TMLR_

### Review · Reviewer_jF2U · 2024-12-06

**Summary Of Contributions:**

This paper conducts a thorough review of the impact of model architecture on the generalization of the features learned by the model (as measured by the performance of these features on other datasets. These "downstream" datasets are drawn from a broad range of domains, and examined as such. The authors find that performance on the pre-training task (imagenet-1K) is not a good predictor of downstream performance, with certain CNN-based architectures, for example ConvNeXt, outperforming Transformers given a fixed parameter count.

**Audience:**

Yes

**Claims And Evidence:**

Yes

**Requested Changes:**

Please address at least the first three weaknesses in the section above. In particular, given the prior knowledge from the papers in the previous section, I am not sure that the authors' findings are very surprising. The authors may wish to change their paper title to something a little closer to the paper's study and conclusions (for instance, the authors could mention that this work compares across architectures).

**Strengths And Weaknesses:**

## Strengths:

1. The paper performs a very thorough analysis, with a large number of architectures compared and a larger number of downstream datasets than one normally sees in these kinds of papers.
2. The use of pre-trained models is quite intelligent, as it allows the authors to focus their resources on downstream training.
3. The breakdown of datasets into broad categories is practically useful
4. The paper is upfront in its focus on real-world utility, which is very appropriate for this work.

## Weaknesses:
1. The background section is very short and fails to mention that the observation that ImageNet peformance does not correlate well with downstream task performance has already been made. In particular, please consider [Kornblith et al. “Do Better ImageNet Models Transfer Better?” CVPR 2019], [Salman et al. "Do Adversarially Robust ImageNet Models Transfer Better?", NeurIPS 2020] and [Iofinova et al. "How Well Do Sparse ImageNet Models Transfer", CVPR 2022]. The last work is especially relevant, since sparsity is also a form of neural architecture search.
2. In light of the above papers, care should be taken that training hyperparameters outside of the architecture, most notably label smoothing, are controlled for; this is not mentioned in the paper.
3. The medical imaging category only consists of two datasets, and so specifying that ConvNext, probably the top choice overall, is unsuitable for this domain seems overly strong.
4. Only ImageNet is used as the backbone dataset. The findings would have been stronger if other pretraining datasets (perhaps there are some domain-specific ones?) were used as well.

---

> ### Author Response · Authors · 2025-01-09
> **Response to Reviewer jF2U**
>
> Thank you for taking the time and effort to review my paper and for providing constructive comments. Your feedback is greatly appreciated and will be invaluable in improving the quality of the work.
>
> We thank the reviewer for pointing out the missing papers which already observed that ImageNet peformance does not correlate well with downstream task performance. We have added these to our Related works section in the revised version.
>
> We have not used label-smoothening for fine-tuning in this work. This is now mentioned in the revised version.
>
> We understand the checking the medical image fine-tuning performance in just 2 datasets is not enough to claim that ConvNeXt is unsuitable in this domain. But since we tested on multiple magnifications of BreakHis, we thought it was a reasonable claim to conclude. We have rewritten the claim in the revised version.
>
> We agree that experimentation using other pre-training datasets would lead to stronger findings. SInce pre-trained weights in other datasets are not available in pytorch, we would require more compute and time to conduct those experiments. We will be working on this in future.
>
> We have changed the title of the paper to "Which Backbone to Use: A Resource-efficient Domain Specific Comparison for Computer Vision".

---

> > ### Comment · Reviewer_jF2U · 2025-01-09
> > **Thank you for your response - clarification on label smoothing**
> >
> > Thank you for your response and for incorporating my and others' feedback; I believe that the paper is stronger for it.
> >
> > One clarification on the question of label smoothing - I was referring not to how the backbones were finetuned, but rather on how they were pretrained (as per the Kornblith paper I referenced earlier). Could you please double check and address this in your manuscript.

---

> > > ### Author Response · Authors · 2025-01-09
> > > **Response to Reviewer jF2U - clarification on label smoothing**
> > >
> > > We did not pre-train the backbones ourselves in ImageNet dataset. We used the pre-trained ImageNet weights available in torchvision library for our fine-tuning experiments for all models, since most practitioners use ImageNet pre-trained weights directly from this library. We do not know how Pytorch Torchvision library pre-trained these models. The official Torchvision github repo code shows that they have added an option for label smoothening in argument parser, but we dont know what value they used. The default value is zero which means no label smoothening.

---

> > > > ### Comment · Reviewer_jF2U · 2025-01-10
> > > >
> > > > It looks like the training hyperparameters are documented here: https://github.com/pytorch/vision/tree/main/references/classification
> > > >
> > > > It should be possible to use this to monitor any deviations from expected defaults and comment on these with reference to the Kornblith work (outside of just label smoothing, which does not appear to have been used)

---

> > > > > ### Author Response · Authors · 2025-01-10
> > > > > **Response to Reviewer jF2U**
> > > > >
> > > > > In Kornblith work, they trained all models on ImageNet with scale parameters for batch normalization layers and without label smoothing, dropout, or auxiliary heads. They found that when fine-tuning on other datasets, regularization and pre-training settings  had little effect upon the performance of fine-tuned models. They noticed that introducing a batch normalization scale parameter and disabling label smoothing improved fine-tuning performance. Also, adding dropout and the auxiliary head sometimes improved performance, but only if used during fine-tuning. They also found that using regularizers at ImageNet pretraining time does not benefit fine-tuning performance unless the same regularizers are used to fine-tune.
> > > > >
> > > > > In Torchvision ImageNet training, amoung the models we compared, only EfficientNetV2, ConvNeXT, Swin, SwinV2, ShuffleNetV2 and WaveMix used label smoothening (with parameter=0.1). We did not employ any label smoothening during fine-tuning. No augmentations were used in ResNet, RegNet and ResNeXT pre-training. Since we did not employ similar augmentations during fine-tuning, we did not observe that models pre-trained without augmentations perform better in this case, verifying the claim made by Kornblith that augmentations during pre-training had negligible impact on fine-tuning performance.

---

### Review · Reviewer_3iRg · 2024-12-10

**Summary Of Contributions:**

The authors provide an analysis of the finetuning performance of various common vision architectures on various datasets. They find that modern architectures like ConvNeXt are more easily finetuned than classic architectures like ResNet, and that such architectures additionally outperform others on natural images in particular.

**Audience:**

Yes

**Claims And Evidence:**

Yes

**Requested Changes:**

It would be nice to include some more experiment details, e.g., if any attempt was made to tune LRs for each model.

**Strengths And Weaknesses:**

Strengths
- Interesting finding that modern architectures perform better when finetuning/there isn't much correlation with ImageNet-1k accuracy and downstream performance
- Tested across many different domains
- I think it's interesting that convolutional models seem to work better on natural images (any speculation as to why?)

Weaknesses
- Unclear if this is a controlled experiment, i.e., maybe the LR/optimizer settings are just better suited for the more modern architectures?
- Attempts to make conclusions about new activation functions/squeeze-excite block from just a few observations

---

> ### Author Response · Authors · 2025-01-09
> **Response to Reviewer 3iRg**
>
> Thank you for your encouraging words and for highlighting the interesting aspects of our work. We truly appreciate your time and effort in reviewing our paper and providing constructive comments. Your insights are invaluable and will certainly help us improve our future work.
>
> Thank you for your insightful comment regarding whether this was a controlled experiment, particularly with respect to the optimizer and learning rate settings. We acknowledge the significance of these factors and have used AdamW for optimization, as it has become a standard for fine-tuning various architectures, including older ones like ResNet. To ensure robustness, we've also employed our own dual optimization scheme, combining AdamW and SGD with momentum. This approach aims to mitigate any discrepancies that might arise from using a single optimizer. As for the learning rate, we have not used any scheduler, relying solely on the optimizer for adaptations. Also, we have not tuned LR for each model. We agree that future experiments are necessary to explore the isolated impact of learning rate and optimizer settings on modern architectures to better understand their contributions.
>
> We have added details in the revised version.

---

### Review · Reviewer_aowS · 2025-01-04

**Summary Of Contributions:**

The paper presents empirical evaluation for transferability of certain architectures (less than 30m parameters) to smaller datasets with fewer than 100k images for the task of image classification.

**Audience:**

Yes

**Claims And Evidence:**

Yes

**Requested Changes:**

Please see the weaknsesses.

**Strengths And Weaknesses:**

Strengths: (1) the paper is crisp and well-written.

(2)The requisite citations for the models being used and their code refs if available are all well-cited.
(3) I think the empirical evaluations are quite extensive and to the best of my knowledge, novel.
(4) the limitations are crisply stated, paving way for future works
(5) presented insights are interesting and actionable.

Weaknesses:  (1) While i dont think the results are known from before, the authors could do a better job of elucidating previous works on transfer learning.
The authors write-“whether these metrics translate to similar relative performance when fine-tuned on smaller niche datasets.” I don’t think this is true. For example, the following paper talks about transfer of regular vs adversarially trained models for smaller datasets. It has 90 citations.
https://arxiv.org/abs/2007.05869

(2) While this is not part of the main contributions of the paper, the authors could do better in writing a bit more about giving a brief introto the various architectures instead of just naming them.

(3) Some of the "insights" towards the end of the paper are redundant. "shufflenet is better than mobilenet" does not add to the exposition as this is obvious from the presented numbers. At the same time, condensed conclusions such as "resnet is consistently overshadowed" is still a welcome insight. I understand this is probably subjective.

(4) I would urge the authors to release their codebase anonymously for now. It is hard to say otherwise if the presented results are reproducible. It seems they have implicitly promised this as there is a sentence saying "code is available at XXXX" but they could just give an anonymized github repo as the link.

Minor:
(maintainers & contributors, 2016) – rewrite the citation ?

---

> ### Author Response · Authors · 2025-01-09
> **Response to Reviewer aowS**
>
> I would like to sincerely thank you for taking the time and effort to review my paper. Your constructive feedback is invaluable, and I truly appreciate the insights you've provided. They will greatly contribute to improving the quality of the paper.
> Thank you for highlighting the omission of certain previous works on transfer learning. We've now included the paper you mentioned in the related sections and revised the sentence to acknowledge this in the revised version.
> We had already written a brief intro to all the architectures used and kept it in the Appendix. We have now moved it to the main paper content in the revised version.
> The original code is already public. But for the review, we have created an anonymous github repo https://github.com/anontmlrsub/tmlr1 which you can use to check if the presented results are reproducible.
> With regard to (maintainers & contributors, 2016) citation for Torchvision, we checked the official github repo of torchvision and they asked to cite as this. We have edited it such that it reads (TorchVision-maintainers & contributors, 2016) now in the revision.

---

### Decision · Action_Editor_HUQE · 2025-02-24

**Recommendation:** Accept as is

**Comment:**

After the authors' revision, most of the reviewers' concerns have been addressed. The reviewers unanimously recommend acceptance of the paper.

I find the work valuable and recommend its acceptance and publication.

**Audience:**

This paper may be of interest to researchers and practitioners exploring the transferability of vision models and the effectiveness of pre-trained models for downstream tasks.

**Claims And Evidence:**

Summary:

This paper presents a comprehensive evaluation of the transferability of lightweight vision architectures (<30M parameters) to smaller datasets (<100K images) for image classification. The study systematically examines the fine-tuning performance of modern architectures, such as ConvNeXt, compared to classic architectures like ResNet, across diverse domains, including natural, medical, deep space, and remote sensing images. Results indicate that CNN-based architectures consistently outperform vision transformers when fine-tuned on limited domain-specific data, challenging the assumption that high performance on large-scale datasets (e.g., ImageNet-1K) translates to superior downstream generalization. Furthermore, controlled comparisons reveal that certain CNN architectures achieve higher performance than others when matched for network size. These findings provide actionable insights for selecting pre-trained backbones in small-scale, proprietary, and niche computer vision applications.

Claims:

The paper makes several key claims: (1) it presents a comprehensive analysis of the fine-tuning performance of lightweight vision models pre-trained on ImageNet across various small-scale downstream classification datasets, revealing no strong correlation between pre-training and downstream performance; (2) CNNs outperform transformer-based architectures when fine-tuned on limited data; (3) certain CNN architectures consistently achieve superior performance in specific scenarios; and (4) architectural innovations in newer models contribute to consistent performance improvements.


Evidence:

The empirical evidence presented in the paper largely supports the claims, and the study's limitations are acknowledged and addressed in the revision.